



# Optimizing Wind Farm Control through Wake Steering using Surrogate Models based on High Fidelity Simulations

Paul Hulsman[1], Søren Juhl Andersen[2], and Tuhfe Göçmen[3]

[1]ForWind – University of Oldenburg, Research Group Wind Energy Systems, Küpkersweg 70, 26129 Oldenburg, Germany

[2]Wind Energy Department, Technical University of Denmark, Anker Engelunds Vej 1, DK-2800 Lyngby, Denmark

[3]Wind Energy Department, Technical University of Denmark, Frederiksborgvej 399, DK-4000 Roskilde, Denmark

**Correspondence:** Søren Juhl Andersen, sjan@dtu.dk

**Abstract.**

This paper aims to develop fast and reliable surrogate models for yaw-based wind farm control. The surrogates, based on polynomial chaos expansion (PCE), are built using high fidelity flow simulations combined with aeroelastic simulations of the turbine performance and loads. Developing a model for wind farm control is a challenging control problem due to the time-varying dynamics of the wake. Both the power output and the loading of the turbines are included in the optimization of wind farm control strategies. Optimization results performed using two Vestas V27 turbines in a row for a specific atmospheric condition suggest that a power gain of almost $3\% \pm 1\%$ can be achieved at close spacing by yawing the upstream turbine more than $15°$. At larger spacing the power gain the optimization shows that yawing is not beneficial as the optimization reverts to normal operation. Furthermore, it was also identified that a reduction of the equivalent loads was obtained at the cost of power production. The total power gains are discussed in relation to the associated model errors and the uncertainty of the surrogate models used in the optimization, and the implication for wind farm control.

## 1 Introduction

Wind farm control technology investigates the interaction between the turbines to optimize the collective operation of the wind farm. Wind turbines operating in the wake of upstream turbine(s) experience power losses and increased fatigue loads. Accordingly, wind farm control aims at providing an increase in power production and a decrease in structural loads while providing better integration of wind power in the grid.

Intentionally yawing the turbine, through the implementation of yaw-based wind-farm control, is the focus of recent studies as one method to reduce the effect of the wake. The objective is to intentionally change the trajectory of the wake in order to increase the power output of the downstream turbines and possibly reduce the fatigue loads.

However, in order to apply yaw-based wind-farm control, a thorough understanding of the wake and the loading on the wind turbine components is necessary. One of the earliest studies investigating the performance of a propeller in yaw has been conducted by Ribner (1943), by Anderson (1979) and by Smulders et al. (1981). Recently the concept of yawing the turbine



has gained renewed interest focusing on the effect of the wake on downstream turbines. The characteristics of the wake have been analyzed in various studies, see for Knudsen et al. (2015) for an overall review. Numerous wind tunnel experiments have shown the potential of redirecting the wake, see *e.g.* Medici and Dahlberg (2003), Medici and Alfredsson (2006) and Bartl et al. (2018). Fleming et al. (2014) evaluated different techniques to redirect the wake using the high-fidelity wind plant simulation tool Simulator for On/Offshore Wind Farm Applications (SOWFA) and has shown that the yaw misalignment has the most promising effect. Gebraad et al. (2017) has shown that this can potentially increase the power output and the annual energy production (AEP) of the total wind farm.

However, these numerical studies are computationally demanding, while dynamic wind farm control requires fast and reliable wake models in the optimization loop. Therefore, engineering models incorporating the effects of wake steering are necessary. One of the most well known models was developed by Jiménez et al. (2010) to describe the wake deflection. This was combined with a multi-zone wake deficit model in FLORIS ((**FLO**w Redirection and **I**nduction in **S**teady-State).

The FLORIS framework, see Annoni et al. (2018), now includes an alternative wake model by Bastankhah and Porté-Agel (2014) and in Bastankhah and Porté-Agel (2016).

Creating a model that combines the prediction of the effect on the power output and the loading effects on the wind turbine components is crucial for the success to apply a yaw based wind farm control. Majority of the studies have only implemented the power gain in the cost function conducted by Gebraad et al. (2016), Fleming et al. (2016), Kragh and Hansen (2015) and Gebraad et al. (2017). Due to the redirection of the wake the fatigue load on the upstream and the downwind turbine could be alleviated. One of the few studies where the structural loads are also reported is van Dijk et al. (2016), in which the FLORIS framework is combined with CCBlade to analyze the impact of yaw control on the power output and the loading. Their results show that a decrease in loading is possible at the reduction of the power production. In Murcia et al. (2018) polynomial surrogate models have been shown to adequately characterize the damage equivalent loads (DEL) and the power production of the turbines.

In this study, we aim to develop and validate yaw-based wind farm control strategies, based on surrogate models through the use of the high fidelity flow solver Ellipsys3D LES and the aeroelastic tool FLEX5. A surrogate model builds a response surfaces based on the input and output of the determined domain. The power and the loading of the turbines estimated by the coupling between Ellipsys3D LES and Flex5 is used to create a domain to train the surrogate model on. The measurement data obtained at the SWiFT facility are used to define the setup of the numerical simulation. Both the power output and the loading of the turbines are to be included in the optimization of wind farm control strategies. In addition, the advantage of using the surrogate model over the conventional analytical models in the optimization is also discussed in this paper.





## 2 Methodology

### 2.1 Simulation of wind turbine wakes and wind turbine response

In this section, the main details regarding the theory and the implementation of EllipSys3D, the rotor modelling, the inflow conditions and the simulation setup for determining the wind turbine wakes and the wind turbine response in this study are

presented. The power and the loads of the wind turbine are calculated by the coupling between EllipSys3D and Flex5. For a more comprehensive description of the coupling between EllipSys3D and Flex5, see, among others, Sørensen et al. (2015)

#### 2.1.1 Large Eddy-Simulation Governing Equation

The large eddy-simulation (LES) was performed with Ellipsys3D, which was developed at the Technical University of Denmark by Michelsen (1992) and by Sørensen (1995). Ellipsys3D is a 3D CFD solver solving the discretized incompressible Navier-

Stokes equation in general curvilinear coordinates using a block-structured finite volume approach described in Sørensen et al. (2015). The large scale turbulence is simulated directly by the Navier-Stokes equations, whereas the turbulence eddies smaller than a predefined grid size, $\delta x$, are modelled using a subgrid-scale model. The incompressible and filtered Navier-Stokes equation is given in Equation 2 and the continuity equation is given in Equation 1. Here, $\bar{u}$ is the filtered velocity, $x$ is the position vector, $P_{\text{Pres.}}$ is the pressure, $\rho$ is the density, $\nu$ is the kinematic viscosity, $f_i$ are external body forces, $t$ is the time

and the notations and $i$ and $j$ are the directional components. The external body forces will be described into more detail in Section 2.1.2, Section 2.1.3 and in Section 2.1.4. During the analysis of the wake behaviour, the more conventional notation for the directional components of the velocity $u$,$v$ and $w$ is used.

$$\frac{\delta u_j}{\delta x_j} = 0 \tag{1}$$

$$\begin{aligned}
\frac{\delta \bar{u}_i}{\delta t} + \frac{\delta \bar{u}_j \bar{u}_i}{\delta x_j} = &-\frac{1}{\rho}\frac{\delta \bar{P}_{\text{Pres.}}}{\delta x_i} + \\
&\frac{\delta}{\delta x_j}((\nu + \nu_{\text{SGS}})(\frac{\delta \bar{u}_i}{\delta x_i} + \frac{\delta \bar{u}_i}{\delta x_i})) + \\
&f_{i,\text{WT}} + f_{i,\text{turb}} + f_{i,\text{PBL}}
\end{aligned} \tag{2}$$

Furthermore, $\nu_{\text{SGS}}$ is the subgrid scale eddy viscosity related to the subgrid stress tensor ($\tau_{ij}$) and the filtered strain tensor ($\bar{S}_{ij}$). The subgrid stress tensor is responsible for the momentum exchange between the subgrid and filtered scales Sørensen et al. (2015). A SGS (Sub Grid Scale) model based on an eddy-viscosity approach is used to model the term $\nu_{\text{SGS}}$. A mixed scale model developed by Ta Phuoc et al. (1994) is used in the LES.

A hybrid scheme formed of a third order QUICK schemes Leonard (1979) and 4th-order central differencing schemes are used to discretize the non-linear terms.





### 2.1.2 Prescribed Boundary Layer

In this study the atmospheric boundary layer is modelled by using body forces ($f_{i,\text{PBL}}$) across the entire computational domain that defines an arbitrary wind shear profile. This gives the possibility to model the shear profile and the atmospheric turbulence independently as described in Troldborg et al. (2014).

### 2.1.3 Atmospheric Turbulence

5 The atmospheric turbulence is introduced to the Navier-Stokes equation by imposing body forces ($f_{i,\text{turb}}$) into the flow applied in a plane upstream of the turbine, see Gilling et al. (2009). The body forces are obtained from the Mann turbulence box described in Mann (1994) and in Mann (1998), which generates a three-dimensional field of the velocity components. The model requires three input parameters: $\alpha_{\text{Turb}} \cdot \epsilon^{2/3}$, $L_{\text{Turb}}$ and $\gamma$, where $\alpha_{\text{Turb}}$ is the Kolmogorov constant, $\epsilon$ is the rate of viscous 10 dissipation of specific turbulent kinetic energy, $L_{\text{Turb}}$ is the turbulence length scale and $\gamma$ is a measure of turbulence anisotropy.

### 2.1.4 Rotor Modelling

Ellipsys3D is coupled with the wind turbine model in the aero-elastic tool Flex5 Øye (1996) using the actuator line (AL) model. The AL model was developed by Sørensen and Shen (2002) and implemented by Mikkelsen (2003) and Troldborg (2009). The AL model calculates and imposes the body forces ($f_{\text{WT}}$) along rotating lines in the numerical domain, and represents the 15 aerodynamic loads of the rotor blades. Flex5 is an aero-elastic tool, which calculates the deflections and load response of the entire turbine. The coupling is performed by transferring the forces calculated with Flex5 to Ellipsys3D described in Sørensen et al. (2015). In each time step the velocities from Ellipsys3D are transferred back to Flex5.

### 2.2 FLORIS Model

FLORIS [1] is one of the most popular frameworks to estimate the wake deficit and the trajectory behind a steered turbine Annoni 20 et al. (2018). Based on the validation of the various models within the FLORIS framework conducted by Annoni et al. (2018) the Gaussian wake model campaign was used for further analysis.

### 2.3 Surrogate Model Theory

The main purpose of a surrogate model is to build a model that defines the relationship between the input and the output of a 25 given data set in a fast and accurate manner. In the recent study of Dimitrov et al. (2018), 6 surrogate models are benchmarked in term of their ability to predict the life-time fatigue loads. The benchmark recommended using the polynomial chaos expansion (PCE) as the most beneficial surrogate method for quick assessment of loads. It is also widely used due to its simplicity and fast convergence in comparison to a full Monte-Carlo simulation, see Murcia et al. (2018) and Sudret (2008). Hence, in this study, the PCE approach is used to create a response surface of the DEL and the power output obtained from Flex5 with

---

[1]The python code of the FLORIS model was obtained from Github and the version '0.1.1' was used. https://github.com/WISDEM/FLORIS





inflow generated by LES..

The surrogate model was built using the python software toolbox Chaospy [2], which is a numerical tool that enables uncertainty quantification using the PCE and the advanced Monte Carlo method. It was first introduced by Feinberg and Langtangen
(2015). Furthermore, the point collocation method is used to build the response surface of the surrogate model in this study. It is a non-intrusive method which builds upon the idea of fitting the polynomials to the input variable by solving a linear system arising from a statistical regression. The response of the surrogate model is build via orthogonal polynomial families using the three term recursion relation, which depends on the selected distribution of the input parameters. For statistically independent distribution of the input parameter, a multivariate polynomial expansion in Chaospy can be created by using the
tensor product rule of uni-variate polynomials described in Feinberg and Langtangen (2015). The standard statistical linear regression approach is used to fit the polynomial sets to the given set of data points, which determines the final coefficients of the surrogate model. The order of the polynomial sets needs to be defined, which influences the output of the surrogate model.

## 2.4 Simulation Setup

High fidelity flow simulations LES are performed in Ellipsys3D with a single turbine for multiple intentional yaw misalignment cases. The angle between the rotor axis and the free-stream velocity is denoted by $\psi$. A positive yaw angle is a clockwise rotation of the turbine and a negative yaw angle is a counter-clockwise rotation viewed from above. A total of 9 simulations are performed with an yaw angles of the upstream turbine of $\psi_1 = -35°, -30°, -25°, -15°, -5°, 0°, 5°, 15°, 30°$. The flow fields are extracted at multiple downstream distances $sx = 3R, 4R, \dots, 18R$, where $R$ is the rotor radius from each simulation. The
flow fields are then used as input to the aero-elastic tool Flex5. The turbines modelled in Flex5 will be referred to as "ghost-turbines" as they do not affect the flow. The "ghost-turbines" are modelled for yaw angles of $\psi_2 = -30°, -27.5°, \dots, 30°$. Table 1 gives an overview of the simulated cases. This created $9 \times 24 \times 30$ combinations of yaw angles for both the upstream and downstream turbine, which are used as input to the surrogate models.

|  | Coupling between Ellipsys3D and Flex5 | Extracted flow field |
|---|---|---|
| $\psi_1$ | $[-35° : 5° : -15°]$, $-5°, 0°, 5°15°, 30°$ | - |
| $\psi_2$ | - | $[-30° : 2.5° : 30°]$ |
| $sx$ | - | $[3R : 1R : 18R]$ |

**Table 1.** Overview of the cases simulated with the coupling between Ellipsys3D LES and Flex5 and the cases simulated with the extracted flow field in Flex5

---

[2]Chaospy: http://chaospy.readthedocs.io/en/master/installation .html#from-source



### 2.4.1 EllipSys3D Setup

The generation of the Mann turbulence box is done with the following paramters: $\alpha_{\text{Turb}} \cdot \epsilon^{2/3} = 0.03379 \, \frac{\text{m}^{4/3}}{\text{s}^2}$, $L_{\text{Turb}} = 156.69 \, \text{m}$ and $\tau = 3.0516$. The prescribed boundary layer is visualized in Figure 1 and the prescribed boundary layer is without any veer.

**Figure 1.** Prescribed boundary layer in EllipSys3D. **Red Area:** Shaded area indicates the wind profile $\pm\sigma$ **Green Area:** Rotor Area

5     The numerical domain used for the simulation with EllipSys3D has $192 \times 920 \times 192$ grid points with 22 cells per blade. Although, the turbine model is simulated without the aerodynamic effects of the wind turbine tower, it still includes the dynamic behaviour of the tower. Every simulation is performed for 1361.62 s ($\approx$ 22 min) after discarding the initial transient as the wake develops behind the turbine. All simulation cases are performed with the same turbulence seed in order to have a direct comparison between the different configurations. Furthermore, the boundary conditions of the numerical domain of the



performed simulation cases are as follows: *No slip* at the Bottom, *Farfield* at the top and *Cyclic Boundary Condition* at the sides.

### 2.4.2 Flex5

A model of the V27 is used for the simulations in Ellipsys3D and for the FLORIS model. The V27 model is based on the V27

5   turbines located at the SWiFT facility as described by Resor and LeBlanc (2014). The main parameters of the V27 wind turbine are given in Table 2, indicating the rated wind speed ($U_\text{rated}$), the rated power ($P_\text{rated}$), the rated rotor speed ($w_\text{rated}$), the rotor radius ($R$), the hub height ($H_\text{Hub}$), the tilt angle ($\theta_\text{Tilt}$) and the tip speed ratio ($\lambda$). The eigenfrequencies are calibrated according to the reported experimental values. Table 3 shows the eigenfrequencies of the V27 model and the experimental values, which are very close.

| Parameters | Value | Unit |
|---|---|---|
| $R$ | 13.5 | [m] |
| $H_\text{Hub}$ | 32.6 | [m] |
| $P_\text{rated}$ | 225 | [kW] |
| $w_\text{rated}$ | 44 | [rpm] |
| $U_\text{rated}$ | 11.74 | [m/s] |
| $\theta_\text{Tilt}$ | 4 | [°] |
| $\lambda$ | 7.61 | [-] |

**Table 2.** Main Parameters of the Vestas V27 given in  Resor and LeBlanc (2014)

| Modal Frequency [Hz] | Experimental | Flex5 model |
|---|---|---|
| 1st rotor flapwise sym. | 2.40 | 2.36 |
| 2nd rotor flapwise sym. | 6.67 | 7.11 |
| 1st tower FA | 1.01 | 0.98 |
| 2nd tower FA | 7.97 | 8.06 |

**Table 3.** Calibration of the V27 model based on the experimental data from  Resor and LeBlanc (2014). 'FA' indicates the fore-aft motion of the turbine.

10   ### 2.4.3 FLORIS Setup and Turbine Re-calibration

The input parameters for FLORIS are determined from the Ellipsys3D setup. Furthermore, the turbine characteristics, shown in Table 2, together with the $C_P$ and $C_T$ curves of the V27 are also used as input. The parameters $P^\text{P}$ and $P^\text{T}$, which characterizes the power loss due to yaw misalignment and tilt angle, are re-calibrated for the V27 using Flex5 under uniform and steady inflow. With the known relation between the power and the yaw, the parameters $P^\text{P}$ and $P^\text{T}$ are determined for the V27 turbine



as 1.4 and 1.25 respectively, with $\eta = 1$. These are lower compared to the values of the NREL 5 MW, which has a value for $P^{\mathrm{P}}$ and $P^{\mathrm{T}}$ equal to 1.88. Note that, the calibration is limited to $P^{\mathrm{T}}$ and $P^{\mathrm{P}}$ for this study, and the default parameters reported in Annoni et al. (2018) are used for wake deflection and velocity deficit estimations.

### 2.4.4 Surrogate Model Setup

Figure 2 gives an overview of the various surrogate models created using the PCE method and the corresponding distribution of the input parameter from which the multivariate polynomials are built. The DEL of the flapwise bending moment and the total bottom tower bending moment are used. This was done in order to simplify the model and to include the most important components influenced by yaw steering. The edgewise bending moment was not included since it is heavily influenced by the gravity.

The surrogate models are constructed using three input parameters, shown on the left side of Figure 2: the upstream yaw angle ($\psi_1$), the downstream yaw angle ($\psi_2$) and the spacing between the upstream and downstream turbine ($sx$). Each input parameter is assigned a distribution, and for the upstream and downstream yaw angle ($\psi_1$ and $\psi_2$) a normal distribution with $\mu = 0°$ and $\sigma = 4.95°$ is used. These values are derived from the distribution of the wind direction at hub height in the LES. A

uniform distribution is used for the turbine spacing, since the downstream distance is fixed at a certain location.

A separate surrogate model was created for the power and the DEL for the upstream and the downstream turbine. The surrogate models built for the power and the DEL of the upstream turbine is only dependant on the upstream yaw angle ($\psi_1$). Whereas, the surrogate models for the downstream turbine are created with the following input: the upstream yaw angle ($\psi_1$), the downstream yaw angle ($\psi_2$) and the spacing between the turbines ($sx$). The output of the models are power, flapwise root

and tower bottom bending moment for both the upstream and downstream turbines, respectively, as shown on the right side in Figure 2.

The models are created with the numerical data between $4R \leq sx \leq 18R$. In addition, the stability of the models are improved by increasing the amount of data points. Multiple realizations are extracted from the time series from Flex5 using a

running mean with a window of $10\,\mathrm{min}$ shifted every $30\,\mathrm{s}$. The same procedure was conducted to determine the DEL of the wind turbine components. In addition, the input variables were normalized for the creation of the surrogate model. The normalization is done in order to give each parameter identical weight, and hence avoid that the model is biased towards one parameter. As a result the output is also normalized, but is converted back to give the absolute values during the analyzes of the results.

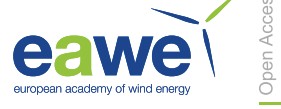
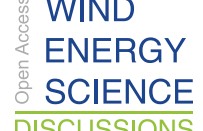

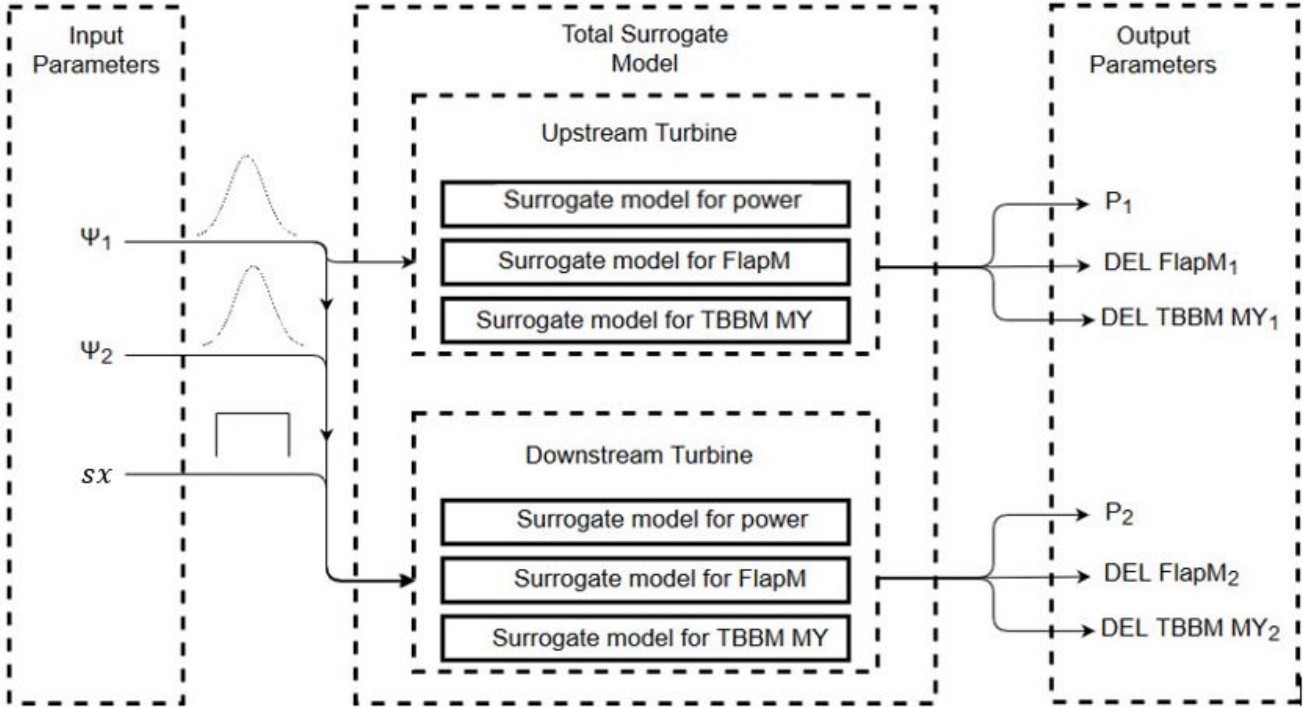

**Figure 2.** Overview of the created surrogate models with the polynomial chaos expansion (PCE) method for the upstream turbine and the downstream turbine. The distribution of the upstream yaw angle ($\psi_1$) and the downstream yaw angle ($\psi_2$) is set to be Gaussian and the spacing between the upstream and the downstream turbine ($sx$) is set to have a uniform distribution. The surrogate model accounts for the power, the DEL of the flapwise bending moment (FlapM) and the DEL of the tower bottom bending moment (TBBM) of the upstream and the downstream turbine

## 3 Results

The turbine performance and the wake deflection is analysed for different yaw angles. The velocity deficit at multiple downstream distances is analyzed and the high fidelity simulations are compared to the FLORIS model. Finally, the surrogate models for the upstream and downstream turbine performance and response are created and used to optimize the operation of the two turbines.

### 3.1 Turbine Performance

The power output of the upstream turbine for $\psi_1 = -30°$, $\psi_1 = 0°$ yaw cases is shown in Figure 3. The power production is initially smooth before the imposed turbulence reach the turbine and the wake starts to develop. The transient period is indicated by the dotted line, and has been discarded from the analysis. The power output is clearly dynamic, and it is evident how the power production of the yawed turbine is significantly lower than normal operation. Furthermore, it is noticeable how

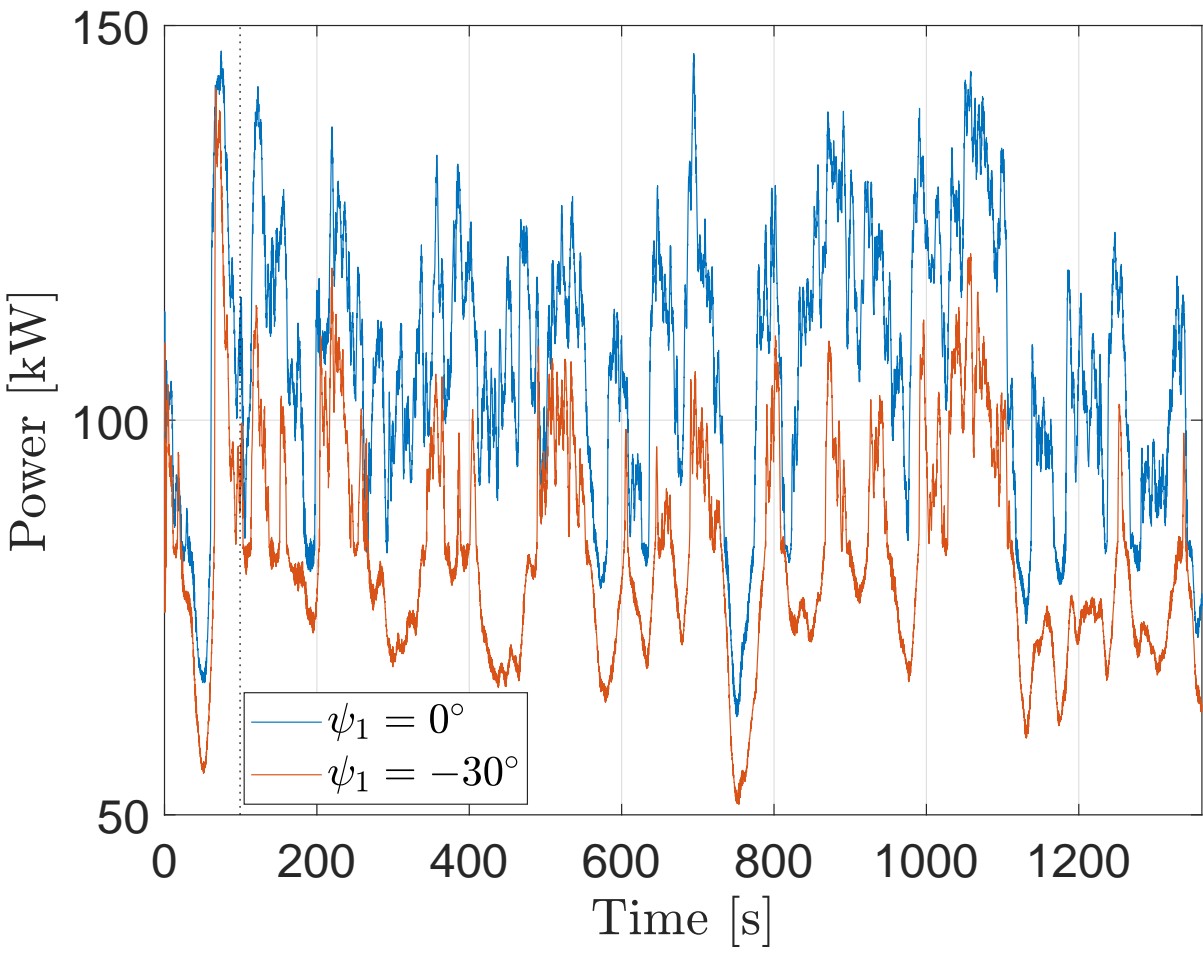

**Figure 3.** Timeseries of the power output of the upstream turbine determined with the aerolastic tool Flex5 at $\psi_1 = 0°$ and at $\psi_1 = -30°$. Dotted lines indicate the end of the transient period

yawing the turbine has a smoothing effect on the power signal, effectively reducing the fluctuations. The smoothing effect increases with increasing yaw angles(not shown for brevity). This interesting phenomena should be further investigated.

## 3.2 Wake Deflection

As the turbine is yawed, the inflow is no longer aligned with the rotor axis. The misalignment leads to a difference in the axial induction and thus an asymmetric loading on the rotor blades, see Branlard (2017) . Additionally, the total thrust also decreases, as demonstrated in  Bastankhah and Porté-Agel (2016).  Bastankhah and Porté-Agel (2016) also illustrated how the wake deflection increases with increasing yaw angle. The wake is deflected as a yawed turbine exerts a lateral force on the



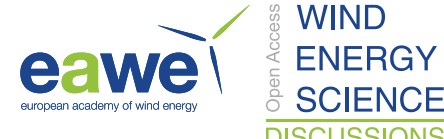

(a) $\psi_1 = 30°$

(b) $\psi_1 = 0°$

(c) $\psi_1 = -30°$

**Figure 4.** Averaged horizontal scan of the averaged relative velocity ($\frac{u}{u_\infty}$) obtained with the coupling between EllipSys3D and Flex5 at $\psi_1 = 30°$ **(a)**, $\psi_1 = 0°$ **(b)** and $\psi_1 = -30°$ **(c)** for the upstream turbine (u is the average streamwise velocity component at hub height, $u_\infty$ is the average incoming streamwise component at hub height)

incoming flow, which induces a spanwise wake velocity due to the conservation of momentum.





The deflected wake is shown in Figure 4 for $\psi_1$ = -30°, $\psi_1$ = 0° and $\psi_1$ = 30°, respectively. The figures show the average relative velocity ($\frac{u}{u_\infty}$) at hub height behind a single turbine. Here, $u_\infty$ is the average incoming streamwise component at hub height. The asymmetric wake is clearly visible. A negative yaw angle leads to a higher deflection compared to a positive yaw angle. The asymmetric wake is also described in Fleming et al. (2018).

### 3.3 Velocity Deficit

The average normalized velocity ($\frac{u}{u_\infty}$) at hub height is shown in Figure 5 for upstream yaw angle of $\psi_1 = 0°$ and at $\psi_1 = -30°$. The results are compared to results from FLORIS.

Clearly, FLORIS is unable to capture the velocity distribution within the near wake region $\forall \frac{sx}{R} \in [0,5]$, but otherwise the results are comparable. In the far wake region, approximately $\forall \frac{sx}{R} \in [5,36]$, the normalized velocity slowly increases as the wake recovers. FLORIS shows a similar behaviour as the high fidelity numerical simulation. Furthermore, Figure 5 indicates the wake center obtained from the FLORIS model and the numerical simulation performed with the coupling between Ellip-Sys3D and Flex5. The wake center is obtained by determining the position of the peak of a fitted two-dimensional Gaussian

distribution. The largest difference in the estimated wake center occurs for large yaw angles. It is also noticeable how the wake center is not perfectly aligned with the rotor axis in the LES results for larger distances due to the relative short averaging time and the effects of shear and turbulence. FLORIS does not account for the asymmetry in the wake deflection, so the comparison is limited to normal operation and $\psi_1 = -30°$. A correction could be implemented on the FLORIS results to account for the asymmetry of the wake deflection. It should also be noted that the overall performance of FLORIS might be improved by re-

calibrating the wake deflection and velocity deficit parameters for this case study. However, the data-driven approach is limited to the surrogate models for this study and the default parameters of FLORIS have been used, as indicated earlier in Section 2.4.3.

### 3.4 Power Surface

The cumulative power production of the upstream turbine and the downstream turbine will be assessed using "ghost-turbines". A "ghost-turbine" is indicated in Figure 4b, where the planes of the velocity components are extracted. The standalone Flex5 simulations are very fast to run and it is possible also to change the yaw angle of the second turbine.

    Figure 6 shows the cumulative power production normalized by the power production of the upstream turbine with no

yaw-misalignment. The cumulative power production is shown as function of downstream yaw angle ($\psi_2$) and the spacing $\frac{sx}{R}$ between the upstream and the downstream turbine for three different upstream yaw angle ($\psi_1$). The power surface shown in Figure 6 are performed with Flex5 simulations conducted at $\psi_1 = -30°$, $0°$, $30°$ with $\psi_2 = -30°, -27.5°, ..., 30°$ and





$sx = 4R, 5R, ..., 18R.$

The cumulative power output increases with an increasing turbine spacing as the wake recovers. Bastankhah and Porté-Agel (2016) showed that an increase in the yaw angle leads to a decrease in the power output, which is also seen in Figure 6. Due

to the reduction of the power output of the downstream turbine, there is a clear reduction of the cumulative power output at large yaw angles of the downstream turbine, $\psi_2$. Furthermore, the effect of wake deflection is clearly visible in Figure 6. The cumulative power output at $\psi_1 = -30°$ and $\psi_1 = 30°$ is higher compared to $\psi_1 = 0°$ at $\psi_2 = -30°, -27.5°, ..., 30°$ and $sx = 4R, 5R, ..., 18R$. The increased power production is due to the wake deflection observed in Figure 4, where a higher velocity reaches the rotor of the downstream turbine. At $\psi_1 = -30°$ the wake is deflected further away, which results in a

higher power output for the downstream wind turbine at closer spacing in comparison to $\psi_1 = 30°$. In addition, the highest power output of the downstream wind turbine is obtained at a slight positive yaw angle for the downstream turbine ($\psi_2$). This is indicated with the red line in Figure 6, showing the yaw angle with the maximum power at a certain turbine spacing. The highest power production is achieved when the wake is approaching the rotor plane perpendicular. For $\psi_1 < 0°$, this results in $\psi_2 > 0°$ and for $\psi_1 > 0°$, this results in $\psi_2 < 0°$. The minor increase in total power production by also yawing the downstream

turbine might not require additional control, but rather be the preferred control setting of the downstream turbine even with a greedy controller as the turbine aims to align itself with the flow automatically.

An increase in the cumulative power production for larger turbine spacing is also seen in the FLORIS results (not shown for brevity). However, as the FLORIS model does not capture the asymmetric behaviour of the wake deflection, the cumulative

power at $\psi_1 = -30°$ is identical to $\psi_1 = 30°$. Furthermore, the effect of a higher power output with a small yaw angle for the downstream turbine is not captured by the FLORIS mode, as FLORIS uses the local free stream velocity described in Gebraad and Van Wingerden (2014).



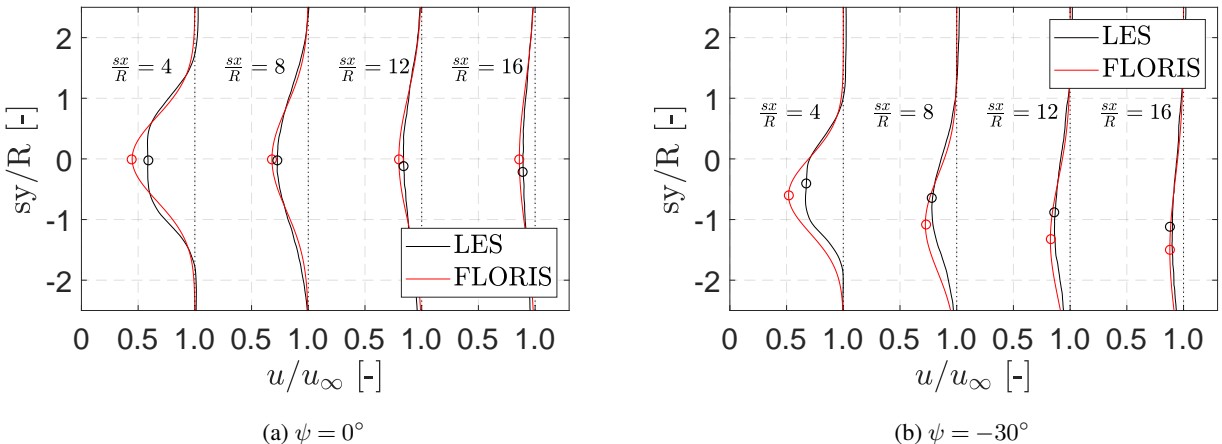

(a) $\psi = 0°$                                    (b) $\psi = -30°$

**Figure 5.** Comparison between the numerical simulation performed in EllipSys3D coupled with Flex5 and the FLORIS model at $\psi_1 = 0°$ and at $\psi_1 = -30°$. The figure illustrates the averaged normalized velocity ($\frac{u}{u_\infty}$) of the wake from the upstream turbine (u indicates the average streamwise velocity component at hub height, $u_\infty$ is the average incoming streamwise velocity component at hub height). **Red Circle:** Indicates the wake center for the FLORIS model. **Black Circle:** Indicates the wake center for LES. The wake center has been determined by fitting the mean wake deficit at hub height to a Gaussian-like function Vollmer et al. (2016)

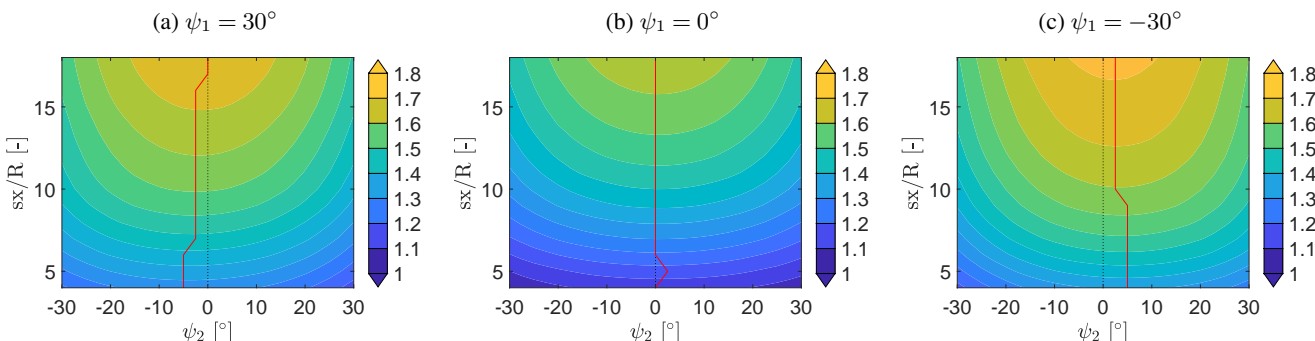

**Figure 6.** Normalized cumulative power of the upstream and downstream turbine determined with the numerical simulation performed in Ellipsys3D coupled with Flex5 at $\psi_1 = -30°, 0°, 30°$ with different spacing and yaw angles for the downstream turbine. The cumulative power is normalized with the power output of the upstream turbine with no yaw-misalignment ($\psi_1 = 0°$). $\frac{sx}{R}$ is the dowsntream distance. $\psi_1$ is the yaw angle of the upstream turbine. $\psi_2$ is the yaw angle of the downstream turbine. **Red Line:** Indicates the yaw angle with the highest power output at each downstream position.

## 3.5 Surrogate Results

The results of the previous sections are used to build surrogate models for the upstream and downstream turbine using PCE as described in Section 2.3 and Section 2.4. The surrogate models are then used in optimization to find potentially beneficial wind



farm control strategies. The analyses of the wake deflection (Section 3.2), the normalized velocity (Section 3.3) and the power output (Section 3.4) will aid the understanding of the outcome of the optimization, with regard to the power and the fatigue loads.

### 3.5.1 Model for the Upstream and the Downstream Turbine

Surrogate models are response surfaces based on a multivariate polynomials. The response surfaces are highly dependent on the selected polynomial order and the training data. In order to visualize the dependency of the response surface with respect to the polynomial order and input data an example case will be used. The example case is based on the power of the downstream turbine at $sx = 5R$ where the upstream turbine yaw angle is fixed. Two surrogate models are built with two different data sets in order to test the sensitivity of the surrogates to the amount of data. Therefore, the first surrogate excludes the data points
at $\psi_1 = -35°$ and $\psi_1 = -25°$ shown in Table 1, while the second surrogate is built using the full data set. In addition, both surrogate models are constructed with a polynomial order of p = 3, 4, 5, 6.

    The response of the surrogate model is illustrated in Figure 7. The black points show the power output obtained from different 10 min periods, which have been extracted using a running mean with a window of 10 min shifted every 30 s over the
time series to generate more samples. Therefore, there are 23 individual points, which are not statistically independent, but gives an indication of the inherent variability. The red point indicates the median value of the simulation point at every unique condition. Figure 7a shows the first power surrogate built on the reduced data set. The surrogate models with p = 5,6 shows signs of over-fitting between $\psi_1 = [-30°, -15°]$ and between $\psi_1 = [15°, 30°]$ for p = 5. This tendency is general throughout the domain, although only shown here for a particular spacing($sx = 5R$) behind the upstream turbine.

    The over-fitting, which results in large errors in the estimates, can be reduced by populating the training data set further. Figure 7b shows the second power surrogate for the full training data set. It highlights how including more data increases the reliability of the fitted polynomial surfaces for all orders of the surrogate models($p = 3, 4, 5, 6$). However, at an order of p = 6 the surrogate model still shows sign of over-fitting between $\psi_1 = [15°, 30°]$. Over-fitting is therefore a significant risk
when fitting higher order polynomials to insufficient data or fairly simple response surfaces and when employing surrogates for extrapolation. On the other hand, it is also seen in Figure 7b that the lower order polynomials is unable to fully capture the deepest normalized power surface around $\psi_1 = \pm 10°$.

    This also indicates that in order to develop a surrogate model based on the simulation data a trade off needs to be made
between an acceptable error, the polynomial order, and the cost of high fidelity simulations. The surrogate models built with the entire data set will be used for the optimization of wind farm control strategies.




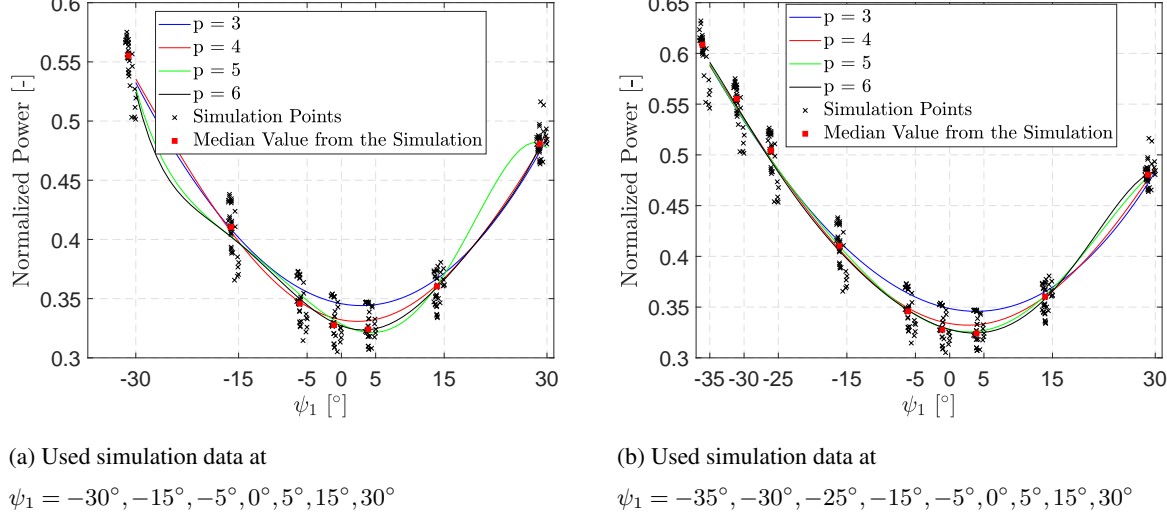

(a) Used simulation data at
$\psi_1 = -30°, -15°, -5°, 0°, 5°, 15°, 30°$

(b) Used simulation data at
$\psi_1 = -35°, -30°, -25°, -15°, -5°, 0°, 5°, 15°, 30°$

**Figure 7.** Power output of the surrogate model of the downstream turbine at $sx = 5R$ and $\psi_2 = 0°$ with two different data sets. The surrogate model has been build using the PCE approach with different polynomial orders (p). **Black:** The crosses indicate the data points obtained by performing a running window of 10 min shifted every 30 s. **Red:** The square indicate the mean of the data points obtained by performing a running window of 10 min shifted every 30 s.

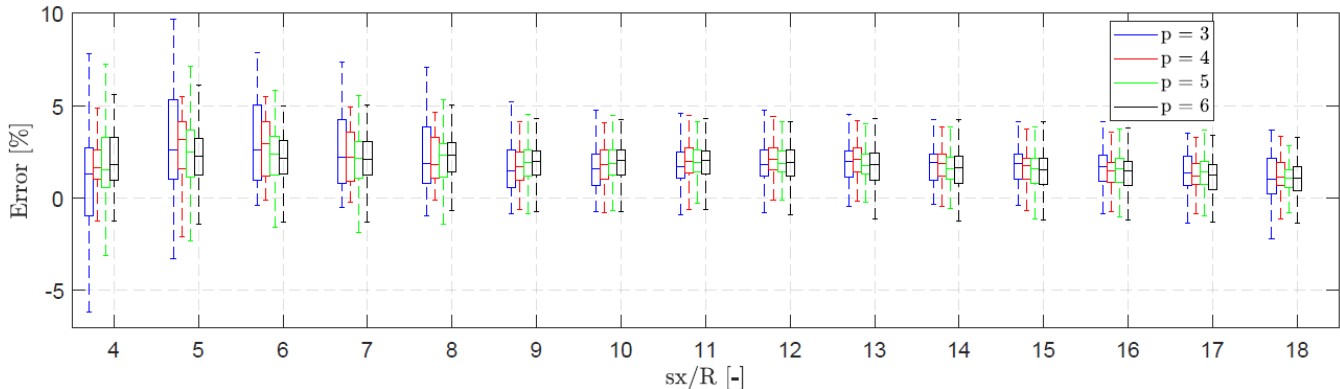

**Figure 8.** Box plots of the relative error, $\xi_R$, of the power output of the downstream turbine between the surrogate model with an order of $p = 3, 4, 5, 6$ and the results obtained from the simulations performed with Flex5, where the flow field is extracted and used as an input to the aeroelastic tool. The distributions corresponds to all the available combinations(225 in total) of the optimization variables $\psi_1$ and $\psi_2$ for each turbine spacing $sx$.





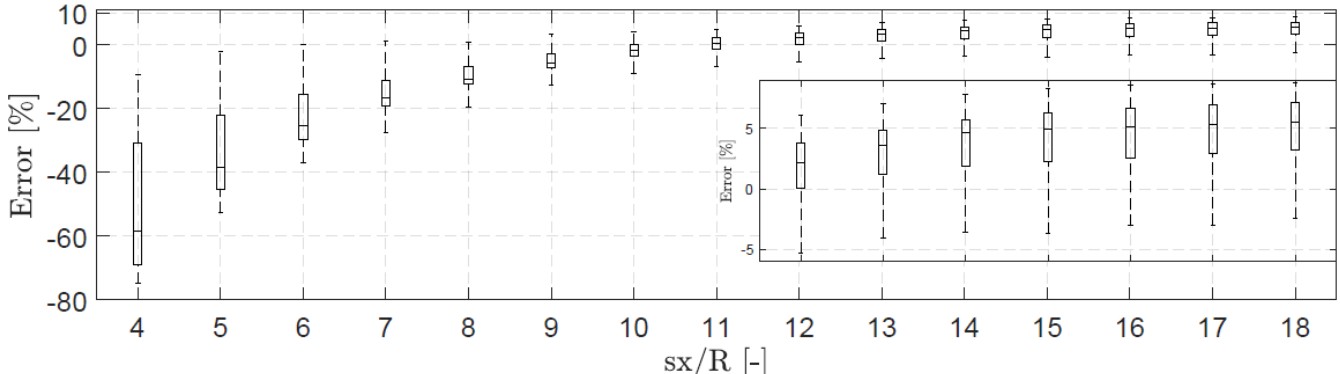

**Figure 9.** Box plots of the relative error of the power output of the downstream turbine between the FLORIS model and the results from the simulations performed with Flex5, where the flow field is extracted and used as an input to the aeroelastic tool. The distributions corresponds to all the available combinations of the optimization variables $\psi_1$ and $\psi_2$ for each turbine spacing $sx$. The insert shows shows the zoom of the last box plots.

### 3.6 Relative Error

The polynomial order for the PCE approach also have to be determined. This choice will be based on the relative error of each surrogate model with different polynomial orders.

5 The relative error is computed for each turbine spacing($\xi_R$) and given by Equation 3.

$$\xi_R(\psi_1, \psi_2, sx) = \frac{P_{SM}(\psi_1, \psi_2, sx) - \widetilde{P}_{Flex5}(\psi_1, \psi_2, sx)}{\widetilde{P}_{Flex5}(\psi_1, \psi_2, sx)} \tag{3}$$

The relative error is the difference between the output of the surrogate model ($P_{SM}$), with various polynomial orders, and the median of the 10-min power output from the simulations performed with Flex5, $\widetilde{P}_{Flex5}$, where the flow field is extracted and used as an input to the aero-elastic tool. The error $\xi_R$ is calculated for the entire domain of the optimization variables, including all the available combinations of $\psi_1$ and $\psi_2$. Therefore, it reflects the overall sensitivity of the model performance to the control set-points. The relative error in the power output of the downstream turbine is shown in Figure 8. Here it can be seen that the distribution of the relative error for all $\psi_1 - \psi_2$ combinations is wider at close spacing for each surrogate model, *i.e.* as the second turbine is within the near wake region (up to $\approx 6R$) of the upstream turbine. At larger spacing, the distribution gets narrower, indicating an increased confidence of the model performance for all $\psi_1 - \psi_2$ combinations for larger turbine spacing.

15 As discussed previously, there is a trade-off between acceptable error and polynomial order, but no decisive conclusion can be drawn with regard to the ideal polynomial order to build the surrogate models. The error distribution of all the polynomial surrogates are comparable and the median relative errors are generally decreased continuously from 3% at $sx = 5R$ to approximately 1% further downstream. Therefore, the surrogate models of polynomial order $p = 3$ and $p = 4$ are used in





the optimization process due to the robustness of the lower orders against over-fitting (as discussed in Section 3.5) and for simplicity. Additionally, it should be noted that the consistent behaviour of $\xi_R$ further downstream indicates a relatively lower sensitivity of the performance of the surrogate models to both the optimization variables $\psi_1$ and $\psi_2$ and to the turbine spacing. This is highly beneficial for a robust optimization process.

The relative error between the simulations performed with Flex5 and the results obtained from the FLORIS model is shown in Figure 9. The relative error is significant within the near wake region, but the relative error reduces further downstream to approximately $5\%$ in the far wake for the present turbine operation and inflow scenario, as also evident in Figure 5. Recent developments of the FLORIS model framework could potentially decrease the relative error, see Martínez-Tossas et al. (2019). However, new additions to FLORIS still relies on model calibration against LES generated data, which could equally be used to increase the population data for building surrogates as previously shown.

### 3.7 Optimization based on the Surrogates

The optimization of wind farm control strategies is performed by applying a weight factor assigned for each surrogate model depending on the objective of the optimization. The aim of this study is to showcase how to develop a control strategy with regard to the loads and the power output through the use of surrogate models. The weighting used for the optimization is shown in Equation 4, where $SM$ is the surrogate model.

$$SM_{\text{TOT}}(\psi_1, \psi_2, S) =$$
$$\begin{bmatrix} n_1 \\ n_2 \\ n_3 \\ n_4 \\ n_5 \end{bmatrix} \begin{bmatrix} SM_{\text{Power, 1}}(\psi_1) + SM_{\text{Power, 2}}(\psi_1, \psi_2, S) \\ 1 - SM_{\text{DEL, FLap, 1}}(\psi_1) \\ 1 - SM_{\text{DEL, FLap, 2}}(\psi_1) \\ 1 - SM_{\text{DEL, TBBM, 1}}(\psi_1) \\ 1 - SM_{\text{DEL, TBBM, 2}}(\psi_1) \end{bmatrix} \qquad (4)$$

Notice, that the weights of the surrogate models for the equivalent loads are subtracted from one since the aim is to reduce the fatigue loads. The optimal point is determined by calculating the power and the equivalent load for the upstream and downstream turbines using the trained surrogates with input parameters of $\psi_1 = [-30° : 0.12° : 30°]$, $\psi_2 = [-30° : 0.12° : 30°]$, and $sx = [4R : 1R : 18R]$. The optimal data point is determined using the standard built-in max-function in python. The optimal control strategy is determined for three cases:

1. a power driven optimization

2. a combined load and power optimization with a large weight attributed to the power

3. a combined load and power optimization with a small weight attributed to the power





### 3.7.1 Power Based Optimization

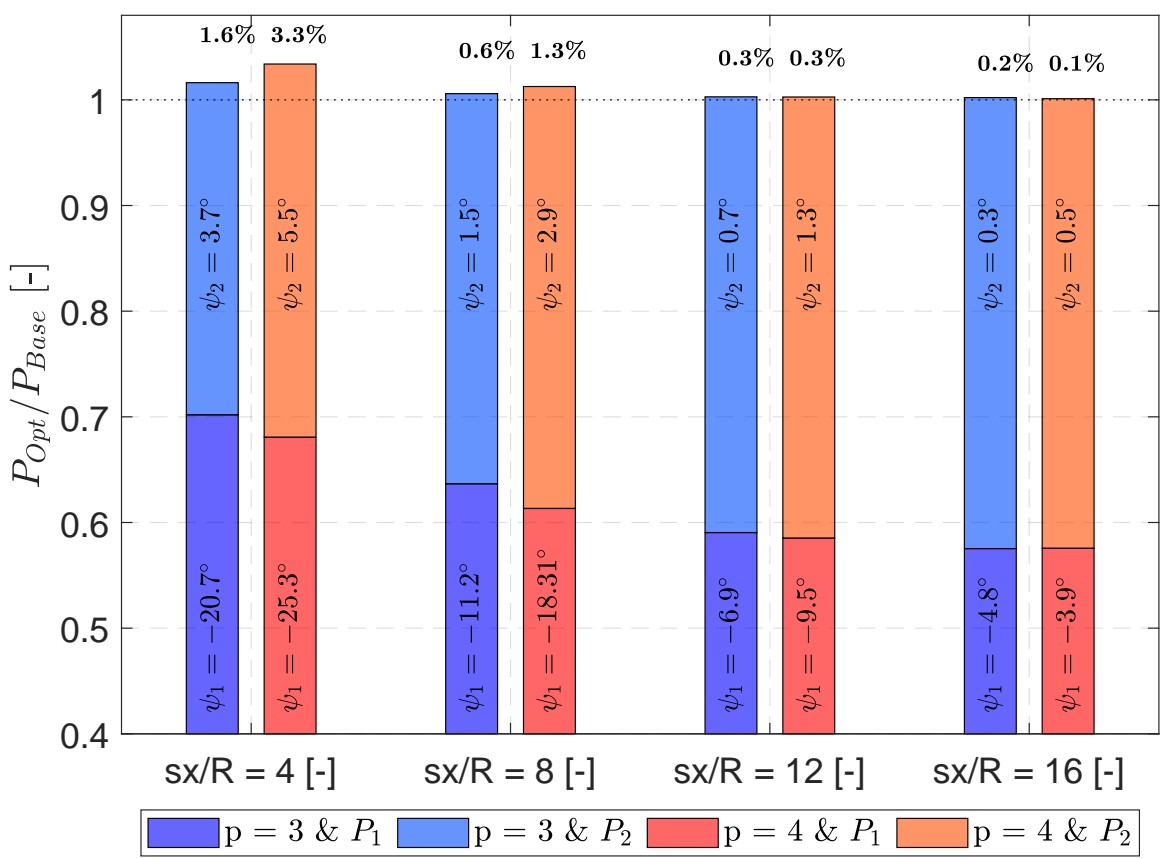

**Figure 10.** Optimal yaw configuration and power gain for a power based optimization. Each bar indicates the cumulative power gain ($P_{Opt}$) normalized with the power of the respective turbine at a certain turbine spacing with no yaw-misalignment ($P_{Base}$), estimated by the surrogate models for both of the turbines. **Darkened:** Normalized power of the upstream turbine **Shaded:** Normalized power of the downstream turbine

The power based optimization is performed using Equation 4 and setting $n_1 = 1$ and $n_2 = n_3 = n_4 = n_5 = 0$ with two surrogate models with an order of $p = 3$ and $p = 4$ for the upstream and downstream turbine. The results of the power based optimization is shown in Figure 10. The figure shows the contribution of the upstream and the downstream turbine to the total

5     power gain, which is the ratio between the estimate power output from the surrogate model compared to the baseline (with $\psi_1 = 0°$ and $\psi_2 = 0°$) power production from the surrogate models.





The figure also depicts the optimal yaw angles of the upstream ($\psi_1 = Opt$)and the downstream ($\psi_2 = Opt$) turbines that maximizes the cumulative power output. Both surrogate models predict that optimal power gain is obtained when the upstream turbine is yawed negatively and the downstream turbine is yawed slightly positively. This is in line with the power output shown in Section 3.4, where the highest power is obtained when the downstream yaw angle ($\psi_2$) is slightly positive. This occurs since

the second turbine aims to reorient itself to be perpendicular with the skewed inflow (see Figure A1 in the Appendix A, which show the average flow angle). It is observed that the power gain is the largest at a turbine spacing of $sx = 4R$ with a power gain of approximately $1.6\%$ and $3.3\%$ for $p = 3$ and $p = 4$, respectively. The difference in estimated power gain is directly related to the overestimation of power production for the baseline of the second turbine in Figure 7 for $p = 3$, which results in the reference $P_{Base}$ being too large. Otherwise, the optimization with both surrogate models of order $p = 3$ and $p = 4$ yield very

similar predictions as the turbine spacing increase. Further downstream, the difference in both power gain and yaw angles of the two turbines are minor.

A negative yaw angle for the upstream turbine is expected as it yields a larger wake deflection, shown in Section 3.2. Furthermore, the optimal yaw angles decrease in magnitude for both upstream and downstream turbine as the turbine spacing

is increased. This occurs as the power loss of the upstream turbine is larger than the power gain of the downstream turbine, which leads to a decrease in the optimal yaw angle of both turbines. However, the uncertainty increase as the yaw angle of the upstream turbine decrease because intentionally yawing the turbine only approximately $-4°$ in the case of $sx = 16R$ is uncertain as the inflow wind direction will continuously change, see *e.g.* Gaumond et al. (2014), and the unintentional yaw misalignment due to erroneous wind direction measurements are often in a similar range.

As shown in Figure 8, the surrogate models come with relative errors as function of the control variables $\psi_1$ and $\psi_2$, which generally indicates an over-prediction of the power production. After the optimization is performed in Figure 10, a closer look to the surrogate model error, $\Delta P_{Opt}$, for each of the optimal control settings is required to correct the model results for potential bias. $\Delta P_{Opt}$ is defined in equation 5 in terms of $\psi_1 = Opt$ and $\psi_2 = Opt$.

$$\mathcal{L}(\Delta P) = \frac{\left(P_{1_{\psi_1=Opt}} + P_{2_{\psi_2=Opt}}\right)_{SM} - \Delta P_{Opt}}{\left(P_{1_{\psi_1=0}} + P_{2_{\psi_2=0}}\right)_{Flex5}} \tag{5}$$

where

$$\Delta P_{Opt} = \left(P_{1_{\psi_1=Opt}} + P_{2_{\psi_2=Opt}}\right)_{SM}$$
$$- \left(P_{1_{\psi_1=Opt}} + P_{2_{\psi_2=Opt}}\right)_{Flex5}$$

Since the dynamic Flex5 simulation gives different results for each 10min realizations (as can been in *e.g.* Figure 7b), the model error also varies among those realizations. Note, that the Flex5 results are not available for all the combinations of $\psi_1$ and $\psi_2$, which is why the surrogate models are built in the first place. However, the prediction error of the surrogates for each

turbine spacing can be interpolated for the optimized yaw angles $\psi_1 = Opt$ and $\psi_2 = Opt$ to estimate $\Delta P_{Opt}$. Median and standard deviation of the model error with respect to the reference Flex5 simulations are presented in Figure 11a as function of turbine spacing.

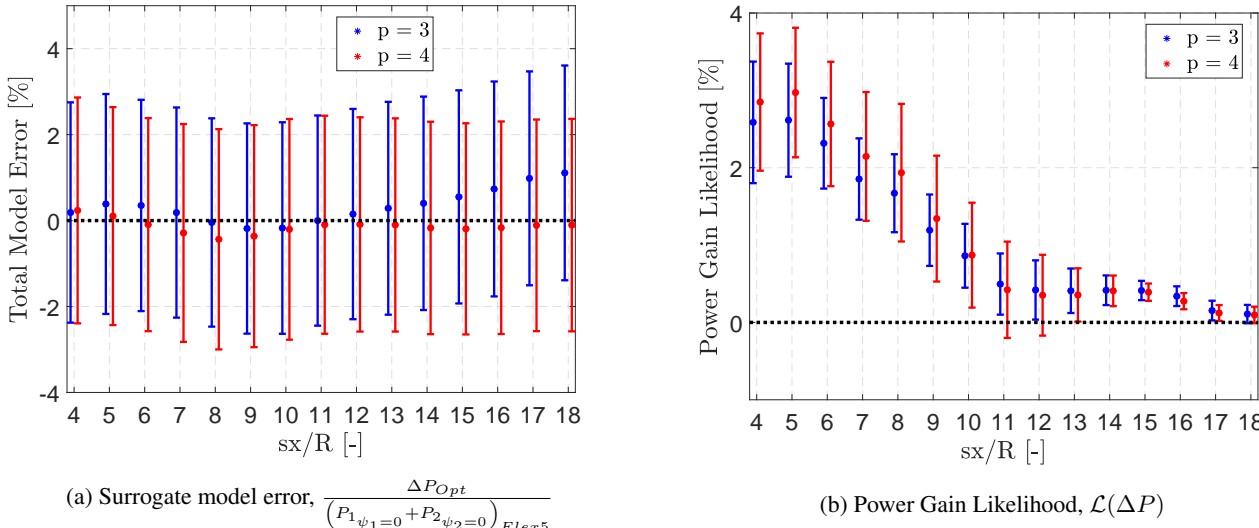

(a) Surrogate model error, $\dfrac{\Delta P_{Opt}}{\left(P_{1_{\psi_1=0}} + P_{2_{\psi_2=0}}\right)_{Flex5}}$

(b) Power Gain Likelihood, $\mathcal{L}(\Delta P)$

**Figure 11.** Surrogate model error and estimated power gain likelihood for the optimized yaw settings $\psi_1$ and $\psi_2$ as function of turbine spacing for PCE model orders $p = 3$ and $p = 4$. Errorbars indicate the median $\pm 1\sigma$ of the difference of the 10-min median realizations compared to the surrogate estimates.

Figure 11a shows that the surrogate models generally reproduce the optimum power very well. For PCE order $p = 3$, the median error increases further downstream, where the optimization settings gets closer to the baseline (*i.e.* $\psi_1 = 0°$ and $\psi_2 = 0°$). This behaviour is expected as it was also observed in Figure 7b for $\psi_1 = 0° \pm 15°$ and $\psi_2 = 0°$. There, it is also seen that the over-prediction is much less visible for PCE order $p = 4$ in that region, which results in much smaller bias for the surrogate especially for larger turbine spacing. The errorbars indicate one standard deviation of the surrogate model error compared to the median of the 23 different 10 min realizations. It shows that the surrogate model error easily both over- and under-predicts by $2\%$, for a specific 10 min realization. Since the variance in the surrogate model error is purely due to the dynamics included in the Flex5 simulations and not the complexity of the surrogate, the spread around the median error for PCE order $p = 3$ and $p = 4$ are essentially the same.

The surrogate model error is subsequently used to assess the power gain likelihood, $\mathcal{L}(\Delta P)$, which is the power estimate including the observed modelling error and uncertainty, as defined in equation 5. The expected power gain likelihood is obtained by subtracting the interpolated surrogate model error (shown in Figure 11a) from the power gain predicted by the surrogate models for each turbine spacing and optimal yaw settings $\left(P_{1_{\psi_1=Opt}} + P_{2_{\psi_2=Opt}}\right)_{SM}$. In other words, due to the potential bias in the model results and difference over the 10-min realizations, implementing the optimum settings indicated in Figure 10 gives different power output, which is a source of uncertainty around the expected power gain by the surrogates. To take this bias and variability into account, the likelihood of the power gain is quantified based on the difference of the surrogate model performance and 10-min realizations of Flex5 simulations as in equation 5 and shown in Figure 11b. It is highly important to Note, that the quantification of the 'power variability' here in this study is limited to the difference between 10-min realizations.





Inter 10-min statistics (or faster time scale investigations) and the ultimate dynamic evaluation of wake steering wind farm control is left as future work.

First and foremost, Figure 11b shows a very similar power gain likelihood for both PCE surrogates $p = 3$ and $p = 4$, also in the near wake; though the results without taking the model error into account were very different as shown in Figure 10.

This shows the importance of the model correction in estimating the *true* power gain to be achieved by the optimized control settings. Note, that in this study, *true* power gain refers to the power gain that is observed in Flex5 simulations for the optimum yaw settings estimated by the surrogate models. For both of the models $p = 3$ and $p = 4$, the power gain likelihood is higher (slightly less than $3\% \pm 1\%$) around the near wake region, where it quickly drops to less than $0.5\%$ around $10R$ downstream. It is also seen that, the standard deviation of the power gain is higher for $p = 4$ which is due to the higher model complexity,

hence higher sensitivity to the model inputs, and higher optimum yaw angles up to $12R$ downstream as presented in Figure 10. Both the expected power gain in Figure 10 (before the model correction) and the power gain likelihood in Figure 11b (after the model correction) points to essentially zero power gain after $10R$ for the investigated setup. It is mainly due to the fact that the optimized yaw setting $\psi_1$ and $\psi_2$ approaches to the baseline case $\psi_1 = 0$ and $\psi_2 = 0$ for larger turbine spacings, *i.e.* no yaw on either turbine. For closer spacings, the wake dynamics are more complex and the associated uncertainty is significant.

The analysis shows that the decision making process to implement the control settings should be based on the power gain likelihood, where the model results are corrected with the best available information, rather than an operational judgement based purely on the model results.

The uncertainty is partly due to the natural variability of the flow and the turbulent wake, and partly due to the uncertainty associated with the surrogate models. The former is inherent and difficult to reduce, while the latter can be reduced by adding

more high fidelity data as shown in Section 3.5.1 and increasing model accuracy. The inherent uncertainties due to the natural variability of the non-stationary atmosphere compared to the current LES model setup is not fully accounted for and atmospheric stability is not included, either. Furthermore, the data used for the surrogates does not take the induction into account, *i.e.* the presence of the turbines alters the inflow and hence the actual power production, see *e.g.* Troldborg and Meyer Forsting (2017) for steady state alterations and Mann et al. (2018) for changes in the incoming turbulent structures.

It should also be noted that wider error distributions and higher bias (seen in Figure 9) in FLORIS would point to wider power gain likelihood distributions, signifying higher risk for the case study. These risks should be quantified and taken into consideration when using *e.g.* FLORIS. Overall, the distribution of the power gain likelihood emphasizes the importance of the uncertainty embedded in the (quasi-)steady models, as well as the mean bias in the error, applied to a dynamic wind farm control setup. In other words, the uncertainty of the model outputs and control inputs has to be taken into account in the decision

making to assess the true performance of the wind farm control strategies. Therefore, applying wake steering for the current turbine and flow scenario might not be a sensible option, except for very small turbine spacing. Even then, the variability and the measurement (or information) uncertainty of the local inflow wind directions might have a significant impact to the power gain likelihood presented here. The challenges of implementing yaw misalignment under uncertainty is briefly discussed in Quick et al. (2017) and should be further investigated for the analyzed setup here in this study.






### 3.7.2 Loads in the Optimization Loop

The loads can be included in the optimization by changing the weight factors ($n_2 - n_5 \neq 0$) assigned to each surrogate model in Equation 4 depending on the objective of the optimization, *i.e.* power or loads.

The power based optimization in the previous section showed that the polynomial order has some influence on the expected power gain, but that it can potentially be corrected for. The DEL of the flapwise root bending moment (FlapM) and the tower bottom bending moment (TBBM) for the upstream and the downstream turbines are surrogated. It should be noted that the total tower bottom bending moment, *i.e.* the total length of the tower bending moment around the $y$-axis and the $z$-axis, have been used here in this study. The combined optimization for both power and loads will be carried out using polynomial of minimum

order, $p = 3$, for each of the load surrogate models to reduce the complexity. In addition, an order of $p = 4$ was selected for the power output of the downstream turbine as it contained smaller total model errors for the expected power gain, see Figure 11a.
Here, the main aim is to visualize the effect of including loads in the optimization process and how it changes the optimum operational condition when applying wind farm control. But it should be noted that the reduction (or increase) in loads under optimum control strategies are not as critical as the power production, because the 'business case' of load reduction is not

as straightforward. For additional information on lifetime extension with regards to load management, see *e.g.* Ziegler et al. (2018)).
As previously shown, weight factors of [1.0, 0.0, 0.0, 0.0, 0.0] would only optimize for the power, while a combined scenario of power and load optimization is generated with two additional weighting factors, namely [0.60, 0.10, 0.10, 0.10, 0.10] and [0.40, 0.15, 0.15, 0.15, 0.15].

Figure 12 shows the results obtained with the different weighting factors for each surrogate model as function of turbine spacing. The results of the upstream turbine are shown with blue-black shades and the downstream turbine in red-orange shades.

Figure 12a shows the optimal yaw angles of the upstream and the downstream turbines. The absolute value of the optimal yaw angles, $\psi_1$ and $\psi_2$, decrease continuously towards 0 for increasing turbine spacing, as previously shown in Figure 10, *i.e.*

yawing the upstream turbine gets less and less beneficial as the turbine spacing increases. The pure power based optimization results in almost no yaw for the largest spacing, while including loads yields larger optimal yaw angles of the first turbine of approximately $-10°$ to $-15°$. Interestingly, included loads in the optimization yields small negative yaw angle of the second turbine, contrary to previous where the second turbine would yaw slightly positive.

Figure 12b shows the power output during the optimization case with different weight factors. Here, it can be seen that the power output reduces with both increasing turbine spacing and increasing weight for the DEL as expected. This is due to the change in the yaw angle, shown in Figure 12a. As the loads are given more weight, the power gain will eventually be a power loss, meaning that power production can be sacrificed to reduce the turbine loads. Furthermore, the green line has been included to directly show the effect of not yawing the second turbine as compared to the optimal yaw settings shown in red.




The additional yawing of the second turbine only has minor influence on the power gain. This is a good sign in terms of the inflow direction uncertainty discussed previously, which would be larger for the second turbine operating in wake.

Figure 12c shows the normalized DEL of the flapwise root bending moment as function of turbine spacing for the different weight factors. Perhaps surprisingly, the loads appear to increase even when including the loads in the optimization for all possible turbine spacing. The DEL increase because the optimization yields a significant reduction in the tower bottom bending moments shown in Figure 12c. It is seen how the tower bottom bending moment can be decreased by almost $20\%$ for the first and $5\%$ for the second turbine, but it comes with a cost of reduced power gain and increased flapwise root bending moment. The transition towards larger turbine spacing is once again continuous.

However, this optimization is of course based on an equal weighting of the flapwise root and tower bottom bending moment. An additional optimization was tested with weights of [0.40, 0.30, 0.30, 0.00, 0.00] and [0.40, 0.00, 0.00, 0.30, 0.30] in order to isolate the effects of only including the flapwise root bending moment and the tower bottom bending moment in the optimization, respectively, as opposed to previous setting where the tower bottom bending moment dominated the optimization space. The results show that it is no longer possible to increase the power production, for such a severe weighting of the loads, see Figure B1 in Appendix B. It is very difficult to decrease the flapwise root bending moments for any of the two turbines, which results in almost no upstream and downstream yaw for the optimization. For the pure tower bottom bending moment, the optimization yields that the first turbine should yaw $-35°$, which is the edge of the training domain for the surrogates, *i.e.* the optimization essentially attempts to turn the turbine out of the wind to reduce the tower bottom bending moment on both turbines.

An improved optimization would require an actual cost model to specify these weights correctly and therefore to assess the economical impact of increasing or decreasing the loads on the different components.

## 3.8  Validation

As any simplified model, the surrogates include model errors and uncertainties as previously mentioned and quantified. Figure 13 shows contours of the normalized power output of the upstream and the downstream turbine with respect to $\psi_1$ and $\psi_2$ at a turbine spacing of $sx = 4R$, obtained from the surrogate models with an order of $p = 3$ and $p = 4$. As seen, the contours differ for the two surrogate models, hence the sensitivity is different. However, the optimal region for both $\psi_1$ and $\psi_2$ is comparable in magnitude, where $p = 4$ yields higher gain for $\psi_2 < -20°$. The extend of the surrogate of order $p = 4$ yields a larger region of higher power increase, as well as a secondary local optima for positive upstream yaw angles. However, it should be noted that around this region, the training data for the surrogate models is sparse hence the confidence to the model results is lower. For the rest of the validation analysis, the turbine spacing of $sx = 4R$ is selected since it provided a high power gain during the optimization of the control strategies.





The validation is performed as a blind test via additional simulations in Flex5 with the optimum yaw settings $\psi_1 = -25.3°$ and $\psi_2 = 5.5°$ estimated by the surrogate model of order $p = 4$ at spacing of $sx = 4R$. Table 4 summarizes the results of the surrogate models and the results using the "ghost" turbines in this Flex5 simulation, which formed the basis for the surrogate models. The surrogate models and the "ghost" turbines yield very comparable results for the power output, as expected. The

5    surrogate models slightly over-estimate the total power output by less than 2%, which is in agreement with the interpolated surrogate model error in Figure 11a. Table 4 also verifies the power gain likelihood, $\mathcal{L}(\Delta P)$ presented in Figure 11b, where the expected value of *true* power gain given by the Flex5 simulations is:

$$\mathbb{E}\left( \frac{\left(P_{1_{\psi_1=Opt}} + P_{2_{\psi_2=Opt}}\right)_{Flex5}}{\left(P_{1_{\psi_1=0}} + P_{2_{\psi_2=0}}\right)_{Flex5}} \right) = 2.1\%, \tag{6}$$

which is less than 0.8 lower than power gain likelihood at $4R$ downstream reported in Figure 11b. Note, that at this spacing,

10   Figure 10 shows the expected power gain estimated by the surrogate models $p = 3$ and $p = 4$ as $1.6\%$ and $3.3\%$, respectively. After taking the model bias and difference in 10min realizations into account, the power gain likelihood of the surrogates have lowered for both $p = 3$ and $p = 4$, approaching to the validation values in Table 4. This once again underlines the importance and added value of model correction in estimating the *true* power gain that is likely to be observed in (quasi-)dynamic operation.

The equivalent load of the flapwise root bending moment determined with the surrogate model also yields very comparable

15   results with the equivalent loads obtained directly from the Flex5 simulations. There is only minor difference in the flapwise root bending moments, while the surrogates underestimate the tower bottom bending moment by less that $5\%$, which is presumably an acceptable difference in terms of loads and which could once again be accounted for by including the model error.

| | $p = 3$ | $p = 4$ | Ghost Turbine Optimum | Ghost Turbine Normal Operation |
|---|---|---|---|---|
| $P_1$ [kW] | 88.3 | 88.7 | 87.3 | 100.9 |
| $P_2$ [kW] | 46.1 | 46.1 | 44.5 | 28.1 |
| $P_{Tot}$ [kW] | 134.4 | 134.8 | 131.8 | 129.0 |
| DEL $FlapM_1$ [kNm] | 25.1 | 25.4 | 26.3 | 23.6 |
| DEL $FlapM_2$ [kNm] | 25.5 | 25.0 | 25.5 | 20.1 |
| DEL $TBBM_1$ [kNm] | 92.4 | 92.2 | 93.9 | 114.6 |
| DEL $TBBM_2$ [kNm] | 158.3 | 161.8 | 165.7 | 106.5 |

**Table 4.** Comparison between the power output and the equivalent loads of the surrogate model for the optimal yaw settings ($\psi_1 = -25.3°$ and $\psi_2 = 5.5°$) and the normal operation ($\psi_1 = 0°$ and $\psi_2 = 0°$) at different polynomial orders for the upstream and the downstream turbine at 4R.





(a) $\psi_1$ and $\psi_2$

(b) Normalized cumulative power production, expected value

(c) Normalized DEL of the flapwise root bending Moment ($FlapM_i$), expected value

(d) Normalized DEL of the Tower Bottom Bending Moment around the,$TBBM_i$

- $\psi_1$/ $FlapM_1$/ $TBBM_1$, $SM_{TOT}(\psi_1,\psi_2,S) = [1, 0, 0, 0, 0]$
- $\psi_1$/ $FlapM_1$/ $TBBM_1$, $SM_{TOT}(\psi_1,\psi_2,S) = [0.60, 0.10, 0.10, 0.10, 0.10]$
- $\psi_1$/ $FlapM_1$/ $TBBM_1$, $SM_{TOT}(\psi_1,\psi_2,S) = [0.40, 0.15, 0.15, 0.15, 0.15]$
- $\psi_2$/ $FlapM_2$/ $TBBM_2$/ $P_1 + P_2$, $SM_{TOT}(\psi_1,\psi_2,S) = [1, 0, 0, 0, 0]$
- $\psi_2$/ $FlapM_2$/ $TBBM_2$/ $P_1 + P_2$, $SM_{TOT}(\psi_1,\psi_2,S) = [0.60, 0.10, 0.10, 0.10, 0.10]$
- $\psi_2$/ $FlapM_2$/ $TBBM_2$/ $P_1 + P_2$, $SM_{TOT}(\psi_1,\psi_2,S) = [0.40, 0.15, 0.15, 0.15, 0.15]$

**Figure 12.** Load based optimization with respect to the DEL of the flapwise root bending moment ($FlapM$) and the DEL of the tower bending moment around the y-axis ($TBBMMY$) for the upstream and the downstream turbine. The weighting is given as follows: [$P_1$ + $P_2$, $FlapM_1$, $FlapM_2$, $TBBMMY_1$, $TBBMMY_2$]. The optimization which included the DEL of the upstream and the downstream turbine was done for three cases. The first case is optimizing only for the power which has the weighting $SM_{TOT} = [1, 0, 0, 0, 0]$. The second case is a combined scenario which optimizes for the power and the load with a weighting of $SM_{TOT} = [0.60, 0.10, 0.10, 0.10, 0.10]$. The final weighting is $SM_{TOT} = [0.4, 0.15, 0.15, 0.15, 0.15]$. **Squared Dots:** Power based optimization. **Circle Dots:** Combined load based optimization **Blue/Black :** $\psi_1$ **Red/Orange :**$\psi_2$. The optimised cumulative power is normalised by the cumulative power during normal operation *i.e.* $\psi_1 = 0°$ and $\psi_2 = 0°$.The DEL are normalised per turbine, as the ratio of the optimum DEL and the DEL during normal operation with $\psi_1 = 0°$ and $\psi_2 = 0°$ for each turbine spacing. **Green:** Normalised cumulative power with $\psi_1 = Opt$ and $\psi_2 = 0°$, to see the effects of downstream yawing.



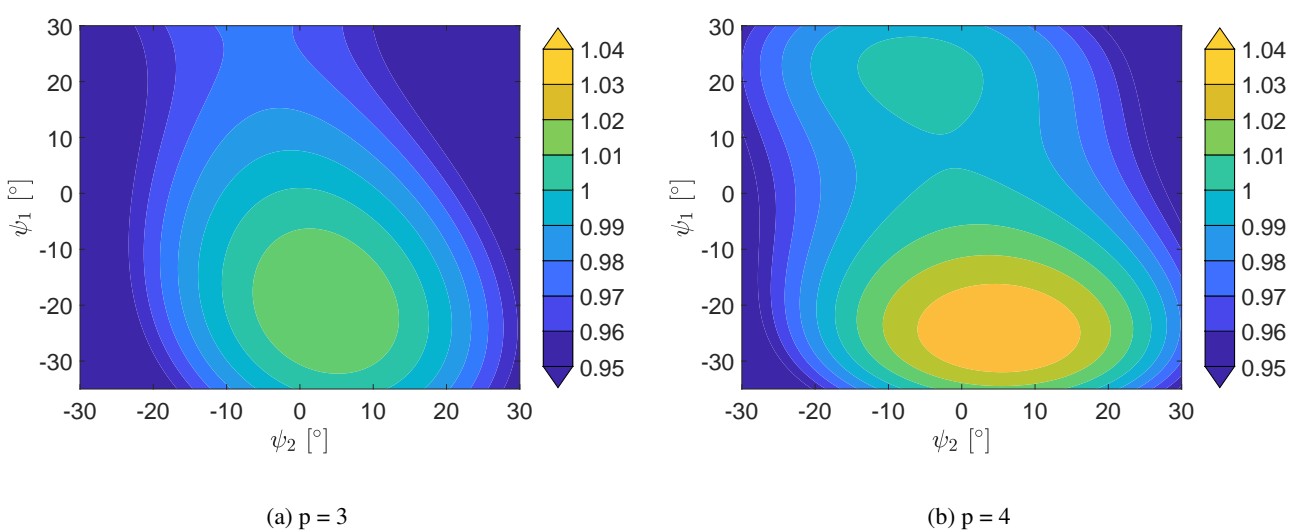

(a) p = 3                                          (b) p = 4

**Figure 13.** Contour plots of the cumulative power output dependant on the upstream and the downstream yaw angle at $sx = 4R$. The power is normalized by the upstream and downstream turbine at $\psi_1/\psi_2 = 0°$



# 4   Conclusion

EllipSys3D has been used to simulate a V27 turbine operating at different yaw angles to investigate wake steering. The turbine
has been modelled using actuator lines, which are fully coupled to the aero-elastic tool, Flex5. The full flow field is extracted
at different downstream distances and input as turbulent field in the standalone Flex5 to mimic a downstream turbine operating
in the deflected wake of an upstream turbine. The performance and response of the two turbines are used to construct surrogate
models based on polynomial chaos expansion.

It is shown how the accuracy of the surrogate models depends on the amount of training data, and how the choice of order for
the polynomials needs to be considered in order to avoid overfitting, yet capture more complexity. The constructed surrogate
models consistently yield median errors for a variety of control inputs, *i.e.* the yaw angles of the upstream and the downstream
turbines at different turbine spacings. Considering the entire domain of the optimization, the surrogate models consistently
over-estimate the power output of the downstream turbine by approximately 2% for most downstream distances. The perfor-
mance of FLORIS is also compared to the high fidelity results for different control settings. FLORIS yields very large relative
errors for close turbine spacings and wide, biased distributions for larger turbine spacings, with median errors of 5% and 3%
standard deviation for the investigated configuration.

Due to their higher accuracy, the surrogate models are used to optimize the power production. The two surrogate models of
order 3 and 4 generally show similar results in terms of total power production of the two turbines, and the optimization leads
to minor yawing (and minor gain) for as the turbine spacing increases. However, the results of the surrogate models with $p = 3$
and $p = 4$ are very different for small spacings due to the difference in the inherit model error and uncertainties. The total power
production is shown to increase up to more than 3% for very close turbine spacing, which decrease to no power gain for larger
distances for the simulated flow scenario. The model error was estimated and found to be very small ($< 1\%$) but with signifi-
cant standard deviation in the total model error of $\pm 2\%$. The optimization results were furthermore corrected by the estimated
model error to give a power gain likelihood, *i.e.* the most realistic optimization performance when correcting for model error
and known uncertainties. However, uncertainties originated from the inherent variability of the inflow and the induction are not
accounted for. The power gain likelihood showed a potential for improving the power production of almost $3\% \pm 1\%$ for turbine
spacings less than $7R$. The power gain likelihood decreased to $0\%$ for larger turbine spacing as the optimization resulted in
only minor or no yaw of the turbines, *i.e.* converging to normal operation as attempting to wake steer becomes unbeneficial.
The associated uncertainty is significant, and the comparison between power gain estimated directly by the surrogate models
and the corrected power gain likelihood emphasize the need to correct model results and take the associated uncertainties into
consideration. In other words, the uncertainty of the model outputs and control inputs has to be considered in order to assess
the true performance of the wind farm control strategies and decide whether to apply a given control strategy. All uncertainties
considered, it might therefore not be sensible to apply wake steering for the current turbine and flow scenario, except if the two





turbines are very closely spaced.

A combination of surrogate models have also been used to include the DEL in the optimization. The results showed that it is possible to reduce the tower bottom bending moments for both turbines by sacrificing some of the power gain. On the other hand, it is generally not possible to reduce the flapwise root bending moments. The combined power and load optimization also tends to converge to normal operation with no yawing of the two turbines for larger spacings.

Finally, the optimization results were compared and validated against additional Flex5 simulation at the optimum yaw angles predicted by the surrogates. The validation confirmed the power gain likelihood assessment and provided estimates of the DEL of both flapwise root and tower bottom bending moments, which were underpredicted by less than $5\%$.

The surrogate approach used in this study could be extended in several ways. To be generally applicable it should include different flow cases *e.g.* wind speed, turbulence intensity, shear, and atmospheric stability. The surrogates can also be expanded by including field measurements when available. Additional surrogate models can be constructed for other turbine models. The true performance test of the presented optimization procedure should be conducted in a wind farm environment, where the flow complexity would increase and hence also the requirements on the model corrections and uncertainty estimations.

*Code and data availability.* The data presented can be made available by contacting the corresponding author.

## Nomenclature

| | |
|---|---|
| AEP | Annual Energy Production |
| AL | Actuator Line |
| CFD | Computational Fluid-Dynamics |
| DEL | Damage Equivalent Load |
| FlapM | Flapwise Root Bending Moment |
| FLORIS | Flow Redirection and Induction in Steady State |
| LES | Large Eddy-Simulation |
| NREL | National Renewable Energy Laboratory |
| PCE | Polynomial Chaos Expansion |
| QUICK | Quadratic Upwind Interpolation for Convective Kinematics |
| SWiFT | Scaled Wind Farm Technology |
| TBBM | Total Tower Bottom Bending Moment |
| TI | Turbulence Intensity |



*Competing interests.* The authors declare that they have no conflict of interest.

*Acknowledgements.* The authors also wish to thank Nikolay Krasimirov Dimitrov for fruitful discussions on the polynomial chaos expansion theory. The study is partially supported by the CONCERT Project (Project no. 2016-1-12396), funded by Energinet.dk under the Public Service Obligation (PSO) and the CCA on Virtual Atmosphere.



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

## Appendix A: Wake Inflow Direction

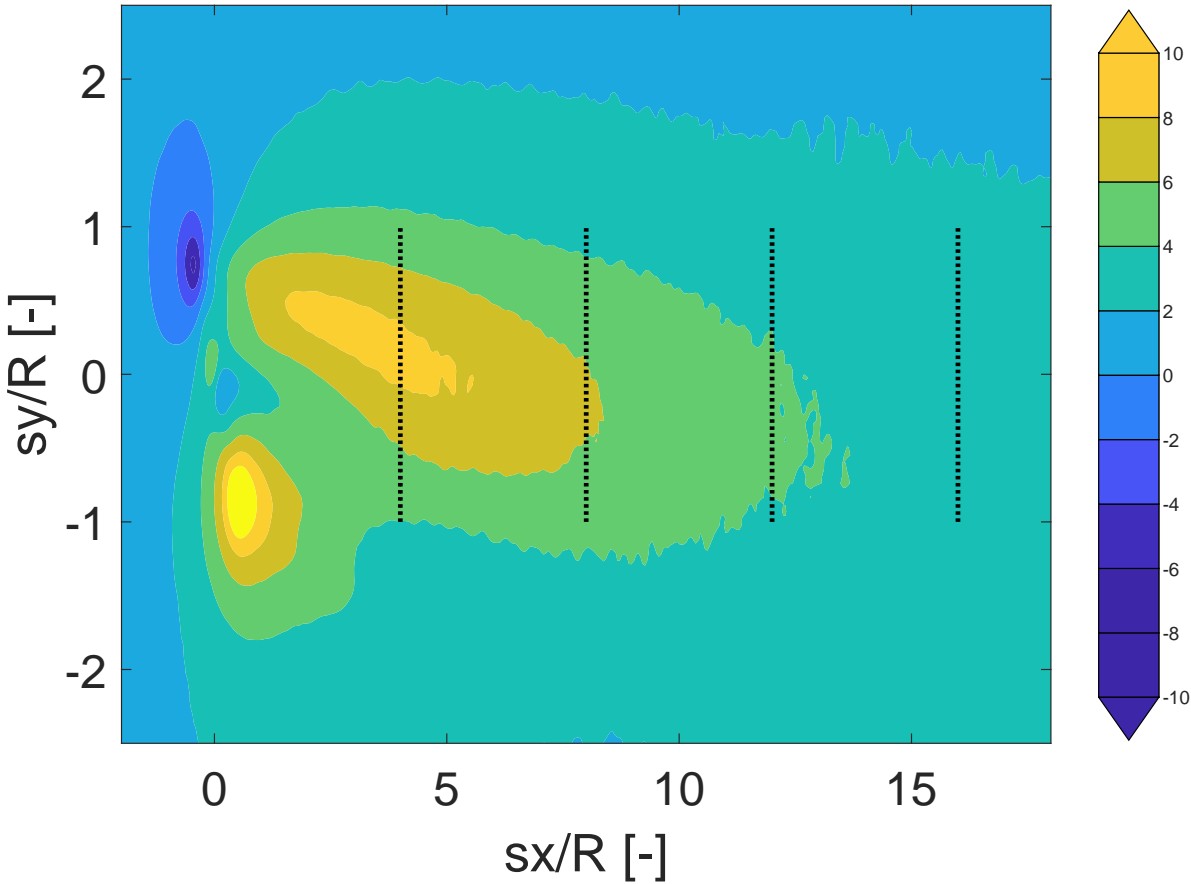

**Figure A1.** Averaged horizontal scan of the flow angle obtained with the coupling between Ellipsys3D and Flex5 at $\psi_1 = -30°$ at hub height. The average flow angle over the rotor area at $\frac{sx}{R} = 4, 8, 12, 16$ is $6.4°$, $5.3°$, $3.8°$ and $3.2°$ respectively. The dotted black lines indicate the rotor area





# Appendix B: Additional Load Based Optimization

(a) $\psi_1$ and $\psi_2$

(b) Normalized cumulative power production, expected value

(c) Normalized DEL of the Flapwise Bending Moment ($FlapM_i$), expected value

(d) Normalized DEL of the Tower Bottom Bending Moment around the, $TBBM_i$

**Figure B1.** Load based optimization with respect to the DEL of the flapwise bending moment ($FlapM$) and the DEL of the tower bending moment ($TBBM$) for the upstream and the downstream turbine. The weighting is given as follows: [$P_1 + P_2$, $FlapM_1$, $FlapM_2$, $TBBMMY_1$, $TBBMMY_2$]. The optimization which included the DEL of the upstream and the downstream turbine was done for two additional cases. The first weighting is $SM_{TOT} = [0.4, 0.3, 0.3, 0, 0]$ The final weighting is $SM_{TOT} = [0.4, 0, 0, 0.3, 0.3]$. The optimised cumulative power is normalised by the cumulative power during normal operation *i.e.* $\psi_1 = 0°$ and $\psi_2 = 0°$. The DEL are normalised per turbine, as the ratio of the optimum DEL and the DEL during normal operation with $\psi_1 = 0°$ and $\psi_2 = 0°$ for each turbine spacing.