# Peer review of "Optimizing Wind Farm Control through Wake Steering using Surrogate Models based on High Fidelity Simulations"

_Wind Energy Science, 2019_

## Referee Comment (RC1) · Anonymous Referee #1 · 16 Sep 2019

The paper looks at the use of approximating physical model results using a surrogate model and then conducting an optimisation. The authors use Ellipsys3D LES AL model coupled to Flex5 and the results are used to build training data for the surrogate model (PCE). The yaw angles are then optimised for power and DEL. The paper is both interesting and novel, the authors provide useful insights into how this approach can be used for turbine yaw control. The paper is very long and although the paper maintains its focus, due to the range of topics, some aspects are covered in only minimal detail. I recommend that the paper should be published with only minor revisions.

The use of PCE is really the key focus of the paper but the model isn't discussed

in much detail. Although references are given, presenting some mathematics for the model in this paper would aid the reader's understanding of what is being done. Make it clear in one place what the parameters of the PCE model are and what inputs are being used.

How does the accuracy of the Ellipsys3D model compare to real measurement data? It seems validation is only conducted for the surrogate against Flex5 and, unless I missed it, no consideration seems to be given for the error of the Ellipsys3D model against reality. Although this is not the focus of the paper, it would give useful context for the results. Consequentially, the use of the phrase 'true power gain', although expanded on in page 22 line 6, should be reconsidered.

The addition of results from FLORIS doesn't add much as it's not part of the Ellipsys3D-Flex5-PCE approach. I can see that it's present as a comparison, but I believe that sections discussing this could be reduced or eliminated without detracting from the paper.

The CFD grid is not described in enough detail and more needs to be presented. What does the grid look like? Why are these grid values used? Was a grid convergence study conducted?

Line 21 page 15 – Relative error is introduced in the following chapter, can an estimate be produced for how large the errors are from overfitting in the case shown in figure 7?

It's mentioned page 22 line 21 that induction is not accounted for. Actually from my understanding of what was done, it would be for the lead turbine as an actuator line is modelled in the fluid domain of Ellipsys3D, but not for the following turbine.

The context for these findings should be presented. How do these results compare to other researchers' findings on yaw control?

Minor and typographical errors:

Equation 2 – Check subscripts in eddy-viscosity terms.

Line 5 page 15, rephrase slightly to be less general.

The sentence at page 20 line 16-19 has some grammatical errors and is hard to follow.

Page 21 line 18 'Note' is capitalised mid-sentence.

Page 25 line 16 'presumably an acceptable difference' is vague and should be reconsidered.

Please check for other spelling and grammatical errors.

———————————————

---

## Referee Comment (RC2) · Anonymous Referee #2 · 1 Oct 2019

The paper studied the use of approximating physical model whose results achieved by the surrogate model. The authors used Ellipsys3D LES AL model coupled to Flex5 whose results are being used to construct training data for the surrogate model. Novelty and approach of this paper are interesting but bit lengthy. Section 2 and 4 have some grammatical and sentence errors. However, the overall paper is acceptable. It would be great if the paper length reduced.

---

## Author Comment (AC1) · 13 Nov 2019

**Reviewer 1:**

The paper looks at the use of approximating physical model results using a surrogate model and then conducting an optimisation. The authors use Ellipsys3D LES AL model coupled to Flex5 and the results are used to build training data for the surrogate model(PCE). The yaw angles are then optimised for power and DEL. The paper is both interesting and novel, the authors provide useful insights into how this approach can be used for turbine yaw control. The paper is very long and although the paper maintains its focus, due to the range of topics, some aspects are covered in only minimal detail.

I recommend that the paper should be published with only minor revisions.

*Thanks for your detailed and positive review. We have addressed all your comments and it has further improved the manuscript.*

The use of PCE is really the key focus of the paper but the model isn't discussed in much detail. Although references are given, presenting some mathematics for the model in this paper would aid the reader's understanding of what is being done. Make it clear in one place what the parameters of the PCE model are and what inputs are being used.

*Thank you for the useful comment, the section assigned to surrogate models have indeed been merged into one (as Section 2.4 now), where a brief mathematical background of PCE is also presented. The parameterization and the inputs used for surrogate models that are described in Figure 2 are further enhanced in Section 2.4.1 Surrogate Training Data set and Section 2.4.2 Surrogate Model Setup. The script of the methodology and an example dataset to train the presented surrogates are uploaded in https://github.com/Paul1994H/PCE-surrogates-for-power-and-loads-under-wind-farm-control.git, as also indicated in the revised version of the article, Section 2.3.2 page 8.*

How does the accuracy of the Ellipsys3D model compare to real measurement data? It seems validation is only conducted for the surrogate against Flex5 and, unless I missed it, no consideration seems to be given for the error of the Ellipsys3D model against reality. Although this is not the focus of the paper, it would give useful context for the results. Consequentially, the use of the phrase 'true power gain', although expanded on in page 22 line 6, should be reconsidered.

*Thanks for the comments. These are very valid points, and we will try to address them here. We have added a few sentences throughout the manuscript, but not as detailed*
*as here as it is not the focus of the paper as also stated by the reviewer.*

*The actuator line is by now a well-established method for high fidelity simulations of wind turbines capable of simulating detailed wake dynamics. Several studies have compared the actuator line method with measurement, see e.g. Troldborg et al. 2010, Shen et al. 2012, and Sørensen et al. 2015. These results show very good comparison with the measurement at different scales, i.e. wind tunnel and full scale.*

*Nevertheless, validation, of course, continues to be an ongoing task. One of the main issues is to actually quantify what is model error and what is model uncertainty. Atmospheric measurements are inherently variable, and even though modelers attempt to match the overall statistics of the inflow, e.g. mean velocity and turbulence profiles, it is extremely difficult to model the exact same flow realization, which leads to model uncertainty, not necessarily a model error.*

*Reversely, this also means that measurements yields very large uncertainties, particular when performing comparative studies with for instance wake steering, where two control settings are examined and attempted to be compared using comparable atmospheric conditions. That requires very long periods of experimental data and statistical handling of the measurements. Here, LES is employed as a virtual experiment, where the exact same flow conditions can be reproduced indefinitely as we alter a single operating condition of the turbine. Thereby, enabling a direct comparison for a single realistic inflow scenario, which would be difficult(=impossible) in atmospheric measurements.*

*Additionally, the model results are compared to the baseline simulation, and in that comparison, the model error is if not constant between simulations, then close to insignificant. So the main issue in terms of model error is only for the comparison with FLORIS, where the LES is assumed to be correct.*

*The input uncertainty, model uncertainty and consequentially the uncertainty in the 'optimum power gain' is thoroughly discussed in the article. Accordingly, the true power gain should also be read in statistical terms – more of psychometric value*

*rather than the reality. Therefore, the explanation in previous version page 22, line 6 is changed from:*
*"Note, that in this study, true power gain refers to the power gain that is observed in Flex5 simulations for the optimum yaw settings estimated by the surrogate models."*
*to:*
*"Note, that in this study, true power gain refers to the power gain that is observed in Flex5 simulations, as an approximation real power gain that one would expect to be able to measure in the field, provided there were sufficient time, money, knowledge of techniques, etc., and no errors in the reporting, collection, and processing of the data"*

The addition of results from FLORIS doesn't add much as it's not part of the Ellipsys3D-Flex5-PCE approach. I can see that it's present as a comparison, but I believe that sections discussing this could be reduced or eliminated without detracting from the paper.

*We have reduced the section on FLORIS significantly and removed a figure. However, we still think the comparison has values. FLORIS have been calibrated against CFD, but if one wishes to perform dynamic wind farm control it should be a fair comparison to apply it on individual 10 min realizations from CFD, and we have added more discussion based on a recent review on wind farm control.*

The CFD grid is not described in enough detail and more needs to be presented. What does the grid look like? Why are these grid values used? Was a grid convergence study conducted?

*The numerical grid has been described in more detail, but we have decided to exclude any graphical representation for brevity. No dedicated grid convergence study was performed, as the actuator line method is well-established and a number of studies*

*have looked into the numerical convergence, e.g. Troldborg 2009 and Forsting et al. 2019. For instance, the later showed a difference of $0.6\%$ in thrust force for a grid resolution of 20 cells per blade. Furthermore, as previously mentioned we here perform a direct comparison of simulation results, so this difference is expected to inconsequential.*

Line 21 page 15 – Relative error is introduced in the following chapter, can an estimate be produced for how large the errors are from overfitting in the case shown in figure 7?

*We have modified the section explaining the relative error. The error from overfitting is given for the surrogate model of the order $p = 5$ and $p = 6$ at $\psi_1 = -25°$, which is $6.4\%$ $7.5\%$ respectively*

It's mentioned page 22 line 21 that induction is not accounted for. Actually from my understanding of what was done, it would be for the lead turbine as an actuator line is modeled in the fluid domain of Ellipsys3D, but not for the following turbine.

*The section regarding the training data set has been re-written in order to make the performed process clearer. The training data for the surrogate models for the upstream and the downstream turbine is obtained from simulations performed with the aero-elastic tool Flex5. This was done to ensure a similar model error/uncertainty between the surrogate of the upstream and downstream turbine, which is particularly critical for the comparison and normalization of the results.*

The context for these findings should be presented. How do these results compare to other researchers' findings on yaw control?

*Good point - now added:*
*"The resulting power gain likelihood presented here can be compared to reported*

*gains in literature, where Kheirabadi and Nagamune (2019) has performed a comprehensive review of the reported power gains published across different model fidelities, wind tunnel, and field tests. The spread in the reported values is very large, ranging from power loss of $-7.9\%$ to power gains of $46\%$ for different wind farm layouts and specifications. However, 16 of the 29 examined studies reported power gains in bins ranging from $-7.5\%$ to $12.5\%$, so the present results are comparable. Furthermore, it is noteworthy how the majority of the studies using low fidelity models, e.g. FLORIS, report power gains of approximately $5\% \pm 2.5\%$, which is comparable to the estimated model error seen in Section 3.4. "*

Minor and typographical errors
Please check for other spelling and grammatical errors

*We have gone through the entire paper and addressed the indicated typographical errors. Furthermore, we have also improved the grammar and sentences.*

**Reviewer 2:** The paper studied the use of approximating physical model whose results achieved by the surrogate model. The authors used Ellipsys3D LES AL model coupled to Flex5 whose results are being used to construct training data for the surrogate model. Novelty and approach of this paper are interesting but bit lengthy. Section 2 and 4 have some grammatical and sentence errors. However, the overall paper is acceptable. It would be great if the paper length reduced.

*Thanks for your positive review. Following the recommendations from the other reviewer, we have tried to reduce the length of the article, particular in terms of the FLORIS comparison. We have also gone through the writing and improved the grammar and sentences.*